# Comparison of Mean Arterial Blood Pressure and Heart Rate Changes in Response to Three Different Randomized Isotonic Crystalloid Boluses in Hypotensive Anesthetized Dogs

**DOI:** 10.3390/ani12141781

**Published:** 2022-07-11

**Authors:** Raphaël Vézina-Audette, Marta Kantyka, Giacomo Gianotti, Deborah C. Silverstein

**Affiliations:** 1Department of Clinical Studies, Matthew J. Ryan Veterinary Hospital, 3900 Delancey Street, University of Pennsylvania, Philadelphia, PA 19104, USA; gianotti@vet.upenn.edu (G.G.); dcsilver@vet.upenn.edu (D.C.S.); 2Section of Anaesthesiology and Pain Therapy, Department of Clinical Veterinary Medicine, Vetsuisse Faculty, University of Bern, 3012 Bern, Switzerland; marta.e.kantyka@gmail.com

**Keywords:** hypotension, fluid therapy, inhalant anesthesia

## Abstract

**Simple Summary:**

Anecdotal reports of a paradoxical exacerbation of hypotension in response to a bolus of a balanced isotonic crystalloid solution containing acetate buffers in dogs under general anesthesia was studied in a prospective clinical trial. The response to three different balanced isotonic crystalloid solutions administered as an intravenous bolus to dogs experiencing hypotension while anesthetized for elective orthopedic or dental procedures was investigated. None of the subjects experienced a paradoxical hypotensive reaction or an increase in blood pressure in response to a fluid bolus.

**Abstract:**

The aim of this prospective, randomized, nonblinded, controlled clinical trial was to compare mean arterial blood pressure (MAP) and heart rate (HR) during an intravenous bolus of three different balanced isotonic crystalloid solutions in euvolemic, anesthetized dogs with hypotension. Thirty healthy dogs (American Society of Anesthesiologists Physical Status I–II) weighing at least 15 kg that presented for elective orthopedic or dental surgical procedures at the Ryan Veterinary Hospital for Small Animals of the University of Pennsylvania were included in this study. Anesthetized hypotensive patients (defined as a MAP ≤ 65 mmHg), were administered an infusion of Lactated Ringer’s solution (LRS), Plasma-Lyte (PLYTE) or Canadian Plasma-Lyte (PLYTECA), selected at random. The infusion was administered over 15 min via a volumetric fluid pump. Differences in oscillometric MAP and HR between time points and across treatments were evaluated by mANOVA. Intravenous isotonic crystalloid infusions over 15 min did not significantly change MAP or HR in hypotensive dogs under general anesthesia. Neither LRS, PLYTE nor PLYTECA exacerbated hypotension or caused tachycardia.

## 1. Introduction

Anecdotal information (personal communication, Baxter International) has been present for years suggesting that rapid infusions of Plasma-Lyte (PLYTE, Baxter International) to hypotensive dogs under general anesthesia could worsen arterial blood pressure and cause tachycardia. Various theories have been brought forward to justify this effect, of which the most common is the cardiovascular effects of rapid infusions of acetate [1] and magnesium (Mg^2+^) [2], both present in PLYTE. However, the literature lacks controlled studies that clearly demonstrate that these components of PLYTE are associated with vasomotor changes during rapid intravenous infusions. Furthermore, studies demonstrating an effect of acetate and Mg^2+^ on vascular tone have used different doses than those associated with clinically relevant crystalloid infusion rates [3,4,5]. Hypotension is a frequent complication observed during general anesthesia in veterinary species [6,7,8,9]. Anesthetic drugs contribute to hypotension by decreasing cardiac output and/or systemic vascular resistance [10]. Hypovolemia can worsen the effects of anesthetic drugs on blood pressure. A recommended initial treatment of perianesthetic hypotension consists of correcting relative or absolute intravascular hypovolemia via administration of intravenous fluids, such as a 10 mL kg^−1^ bolus over 15 min [6]. However, although cardiac output increases following IV fluid administration in both euvolemic and hypovolemic patients, the blood pressure of euvolemic patients under anesthesia is not consistently improved by rapid administration of high volumes of IV fluids [11,12]. In a previous study assessing the effects of IV infusions of Lactated Ringer’s solution (LRS) at 0, 10 and 20 mL kg^−1^ r^−1^ to euvolemic, normotensive, isoflurane-anaesthetized dogs, no significant changes in HR and blood pressures were found between groups after 30 and 60 min of fluid therapy [13]. Differences in the composition of isotonic crystalloid solutions may also contribute to this variability, but has not been studied in dogs.

Crystalloid solutions contain solutes of low molecular weight (<500 g/mol) [14]. They are devoid of large molecules and, consequently, do not have colloidal effects. Different isotonic crystalloid fluid formulations are commercially available in veterinary and human medicine, each varying significantly in their electrolyte concentrations, osmolarity and buffering capacity. Although previous studies have compared the efficiency of crystalloids and colloid solutions at improving blood pressure in anesthetized dogs [15], there is no scientific evidence suggesting the superiority of any fluid type for correction of intra-anesthetic hypotension in dogs. The formulation of Canadian Plasma-Lyte (PLYTECA–Baxter Canada), available for the Canadian market, differs from the American formulation of PLYTE via its decreased Mg^2+^ content (150 nM vs. 300 nM, respectively). Both are otherwise identical in their pH, electrolyte compositions and osmolarity (294 mOsm L^−1^). The composition of the three crystalloids used in this study is outlined in Table 1.

The aim of this study was to compare the effects of three different isotonic crystalloids (LRS (Braun Medical Inc.), PLYTE and PLYTECA) administered as an intravenous bolus on MAP and HR during anesthesia-induced hypotension in dogs and assess the contribution of ionic buffers (acetate vs. lactate) or Mg^2+^ concentration to observed changes in MAP and HR. We hypothesized that the administration of PLYTE as an IV bolus is associated with a higher incidence or magnitude of hypotension and tachycardia than PLYTECA, and that LRS, which contains no supplemental Mg^2+^ nor acetate, but instead contains Ca^2+^, would lead to an increase in MAP and HR.

## 2. Materials and Methods

Study design

Prospective, randomized, nonblinded, controlled clinical trial.

Animals

The study was approved by the University of Pennsylvania’s Institutional Animal Care and Use Committee. Thirty adult dogs were included in this study. All dogs were client owned and anesthetized for elective orthopedic or dental surgery at the Ryan Veterinary Hospital for Small Animals. The inclusion criteria were body weight > 15 kg, at least 6 months of age, no known systemic illness, normal physical examination with the exception of a surgical orthopedic condition or mild dental disease, and complete blood count, packed cell volume, total solids and serum biochemical profile within the reference range.

Anesthesia

All dogs received 0.01 mg kg^−1^ IV or 0.02 mg kg^−1^ IM acepromazine (Aceproject, Boehringer Ingelheim Vetmedica, Inc. St-Joseph, MO, USA) and 0.03 mg kg^−1^ IV or 0.05 mg kg^−1^ IM oxymorphone (DSM pharmaceuticals) for premedication 15 to 30 min before aseptic placement of an IV catheter (BD Insyte, Sandy) in a cephalic vein. Anesthesia was induced with propofol (Diprivan, Fresenius Kabi) administered to effect and maintained with isoflurane at a concentration needed to maintain an adequate depth of anesthesia (vaporizer dial range 1.5–2.5%), as judged by an experienced anesthetist. All patients were maintained on spontaneous ventilation during the recording period. Regional anesthesia was included as part of balanced anesthetic protocols and varied (epidural vs. radial, ulnar, median and musculocutaneous block vs. dental blocks) depending on the procedure. An IV infusion of LRS was administered at a rate of 2–5 mL kg^−1^ r^−1^ by means of a volumetric fluid pump (Hospira Micro Macro Plum XL IV Infusion Pump, Hospira), starting at induction and maintained throughout anesthesia. Antibiotics (cefazolin), if indicated, were administered slowly intravenously for antimicrobial prophylaxis after induction and repeated every 90 min thereafter until closure of the surgical site. After endotracheal intubation, heart rate (HR) was monitored with a lead II ECG, with gel-coated ECG pads (Ambu A/S) affixed to the right and left front paws and left hind paw, oxygen saturation was monitored by means of pulse oximetry and mean blood pressure was measured by oscillometry with a multi-parameter and anesthetic agent monitor (Model S/5 Compact, Datex-Ohmeda; GE Healthcare). For oscillometric blood pressure measurement, an inflatable cuff (Critikon NIBP inflatable cuff, GE Healthcare), the width of which measured 40% of the circumference of the limb, was used. Cuffs were placed either on the thoracic limbs, just distal to the elbows or on pelvic limbs proximal to the tarsus. Although the site of blood pressure measurement was not standardized between patients, it remained constant during the entire recording period for each individual patient. All vital parameters were measured and recorded every 5 min, except body temperature which was measured rectally with a digital thermometer (model: 524038; BD Consumer Healthcare) at 15 min intervals. Temperature was maintained above 37.2 °C by means of circulating warm water beds placed between the operating tables and the patient, and a Bair Hugger (Bair Hugger, 3M) that provided forced warm-air flow to the patient.

Following two consecutive MAP measurements less than or equal to 65 mmHg taken at 5 min intervals (time points −5 and 0) prior to onset of surgery, a third party then performed randomization using the “randbetween” function (1–3, corresponding to LRS, PLYTE or PLYTECA, respectively) with personal computer software (Microsoft Excel, Microsoft Office 2007, Microsoft Corp). Ultimately, patients were administered the selected crystalloid as an IV bolus over 15 min with a volumetric infusion pump set at 999 mL/h for patients over 25 kg, or 10 mL per kg over 15 min for patients weighing less than 25 kg. The fluid rates used to deliver fluid boluses to each subject was limited by the maximum rate of the pumps (999 mL r^−1^). Thus, patients weighing 25 kg and above received a maximum of 250 mL over a 15 min period. Fluid rates administered to patients less than 25 kg were limited to 10 mL kg^−1^ over 15 min. Baseline physiological values (Time point 0) were recorded just prior to the start of the infusion. The anesthetist was not blinded to the fluid type assigned to the patient. During the course of the treatment, the anesthetic depth (inhalant vaporizer setting) was kept unchanged unless the anesthetic plane was too light for the procedure in progress (surgical clipping, scrub, neuraxial or regional analgesia technique). Start of surgery did not occur in any case before completion of the fluid bolus to minimize the effect of surgical stimulation on the vital parameters studied. The study was terminated at the end of the infusion regardless of whether normotension was established by the end of the intervention. If hypotension persisted, standard anesthetic procedures in place at this institution were followed to achieve normal blood pressure.

Statistical analysis

All statistical analyses were performed using a commercial statistical software (Minitab 17, Minitab Inc.). Normality of data distribution was assessed for all variables using the Anderson–Darling test. Repeated-measures 2-way ANOVA with Bonferroni post hoc comparisons were performed with the general linear model tool for normally distributed data and medians of non-normally distributed variables were compared between groups by means of Mood’s median test. Despite the small number of patients included in this study (*n* = 30), a statistical analysis suggested that the sample size was sufficient to detect changes in blood pressure and HR equivalent to 20% and 35% from baseline, respectively, when the alpha = 0.05 and at 80% power.

## 3. Results

### 3.1. Distribution Statistics and Normality Tests

Between August 2015 and March 2016, a total of 30 adult dogs (16 males and 14 females), with ages of 8 months to 8 years (median 2.5 years old), were recruited in this study and were randomly assigned to three treatment groups: LRS (*n* = 13), PLYTE (*n* = 9) and PLYTECA (*n* = 8). Age was not normally distributed (*p* < 0.005). Recruitment ceased once the target *n* = 30 was reached. Although weights were normally distributed (mean 33 ± 10 kg, range: 16 to 53 kg) amongst all subjects, the mean body weight in the LRS group was significantly smaller than in the PLYTECA group (*p* = 0.049) (Table 2). The mean propofol dose required for induction of anesthesia was 2.7 ± 1.8 mg kg^−1^. The mean fluid rate administered to subjects in the LRS group (8.5 mL kg^−1^) was significantly greater than in the PLYTECA group (6.8 mL kg^−1^) (*p* = 0.048). No adverse effect was observed in any group as a result of the infusions.

### 3.2. Comparison of MAP and HR of Dogs during Fluid Bolus

The MAP and HR values for all subjects at each time point during each treatment were normally distributed. The mean time to onset of hypotension occurred at 25 min (LRS), 26 min (PLYTE) and 29 min (PLYTECA).

Following the onset of hypotension, the administration of either LRS, PLYTE or PLYTECA as a rapid intravenous infusion did not result in a statistically significant change in MAP at 5, 10, 15 or 20 min (*p* > 0.05) (Table 3). Furthermore, neither treatment caused a significant change in mean HR at any time over the course of the fluid bolus (*p* > 0.05). Mean body temperatures were within normal limits at the start of treatment and did not differ significantly between groups (LRS: T = 37.3 °C; PLYTE: T = 37.4 °C; PLYTECA: T = 37.4 °C). No adverse effect was observed in any group as a result of the infusions.

Despite the lack of significant effect of treatments on MAP and HR over the treatment period, in the present study, one patient in the PLYTE, three patients in the PLYTECA and two patients in the LRS group underwent a decrease in MAP from baseline after 5 min of administration of a fluid bolus, with small and variable increases in HR. One patient in the PLYTECA group underwent a decrease in MAP between t = 5 min and t = 10 min. Four patients in the PLYTE group had increases in MAP from below to above threshold for hypotension within 5 min of administration of a 10 mL kg^−1^ bolus, whereas three patients in the PLYTECA group underwent similar increases in blood pressure between t = 5 min and t = 10 min. In the LRS group, only two patients had increased MAP compared to baseline values, and both occurred only at t = 20 min.

## 4. Discussion

In agreement with previous studies investigating the effects of isotonic crystalloid fluids in anesthetized dogs [11,12,16], the administration of a rapid infusion of isotonic crystalloid solutions in euvolemic, hypotensive anesthetized dogs did not lead to significant changes in MAP or HR. Further, the data hereby presented suggests that despite differences in the osmolarity of electrolyte and buffering compositions, the choice of LRS, PLYTE and PLYTECA is not associated with significant differences in MAP and HR after fast IV administration to isoflurane-anesthetized hypotensive dogs. The results of this study may not extend to patients with systemic illness or hypotension of different etiology.

Anecdotally, administration of PLYTE infusions has resulted in a paradoxical response characterized by transient hypotension and tachycardia, whereas PLYTECA and LRS are not associated with this response. LRS is another isotonic (273 mOsm L^−1^) crystalloid solution that contains lactate as a buffer instead of acetate. Of note, in contrast with PLYTE and PLYTECA, LRS does not contain supplemental Mg^2+^, but instead is supplemented with 3 mEq L^−1^ of calcium. LRS has not been associated with the occurrence of paradoxical cardiovascular responses in dogs. Hypotension following the rapid IV administration of balanced isotonic crystalloid solutions may be explained by the ionic or buffering molecules in the solution: increases in Mg^2+^ can cause vasodilation as it inhibits smooth muscle contraction. Although mechanisms of Mg^2+^-induced vasodilation are not fully understood, it has been postulated that Mg^2+^ directly competes with extracellular calcium (Ca^2+^) signaling on the surface of vascular smooth muscle. The involvement of endothelial nitrous oxide as a result of increased ionized magnesium concentration has also been suggested [17]. Ultimately, Mg^2+^ administration may lead to hypotension [2]. However, serial serum magnesium levels were not analyzed and the difference in magnesium dose administered during a fluid bolus may not be sufficient to effect a significant change in a patient’s serum concentration, let alone their blood pressure. Calcium ions also act on smooth muscles, but conversely cause vasoconstriction [18,19]. Moreover, acetate anions can also cause a decrease in systemic vascular resistance by inducing a mild vasodilation [1]. Adenosine has been suggested as a key intracellular signaling molecule involved in acetate-induced vasodilation [20], but an adenosine independent mechanism of acetate-induced vasodilation resulting in smooth muscle relaxation and accumulation of cyclic AMP has also been described [21].

Following a rapid IV fluid infusion, intravascular volume is increased. The subsequent increase in venous return causes activation of right atrial stretch receptors, triggering an increase in HR. This reflex, named the Bainbridge reflex, may explain the occasional tachycardia observed in dogs following the administration of a rapid IV fluid bolus [22]. This theory does not explain why this response would be preferentially observed with one crystalloid and not another. The present results suggest that differences in isotonic crystalloid fluid compositions are unlikely to explain the rare occurrence of worsened hypotension and accompanying tachycardia in anesthetized hypotensive dogs receiving rapid IV fluid infusions.

Although this study was prospective and randomized, it was limited by its observational nature and that the complete standardization of the individual procedures in the clinical setting in which data was acquired was not possible. Some patients presented for orthopedic procedures and received epidural morphine and local anesthetics, which can cause hypotension in dogs secondary to vasodilation resulting from a sympathetic blockade. The pharmacological mechanism for the development of hypotension associated with neuro-axial anesthesia is different than that observed in patients maintained strictly under inhalation anesthesia and may exacerbate hypotension and decrease the responsiveness to IV fluid infusion [23,24]. Variations in the timing of events may dilute the effects observed on individual patients when evaluating differences between groups. For instance, the delay between time of premedication and time of induction of anesthesia, or onset of hypotension between patients introduces a source of variation. Specifically, variable delays between administration of premedication drugs and onset of hypotension may be associated with different plasma levels of acepromazine in patients and, thus, different effects on the cardiovascular system. The variability in plasma levels of vasoactive drugs may also stem from the dosage variation and lack of standardized route of administration. However, retrospective analysis of delays between premedication or induction of anesthesia and onset of hypotension in the current experiment shows little overall variability in timing, with a few exceptions considered as outliers. Some patients were presented to the anesthesia department with IV catheters already in place and, therefore, received all drugs intravenously. As a result, drug dosages were decreased relative to patients who required IM sedation prior to placement of IV catheters.

Importantly, we detected a statistically significant difference in the weight of dogs between the LRS and the PLYTECA group. Fluid rates were calculated based on patient weights, but were also limited by the maximum fluid rate of the pumps that were used (999 mL h^−1^). Although the experimental protocol prescribed a 10 mL kg^−1^ fluid infusion over 15 min, patients weighing more than 25 kg received less than 10 mL kg^−1^ over the 15 min period. This may account for the smaller, albeit not statistically significant, median % change in MAP between the PLYTECA group compared to the LRS group.

Although it was specified in the experimental protocol that inhalant levels were to be maintained at a fixed level for the duration of data acquisition, the exact levels of anesthetics required to keep the patient at an adequate plane of anesthesia may have varied considerably between subjects. Additionally, the end tidal inhalant levels were not recorded nor controlled. It is possible that the depth of anesthesia may have been a more important contributor to the continued hypotension in the subjects of this study, rather than the initiation of a rapid IV fluid infusion.

Blood pressure monitoring was performed indirectly via oscillometric BP monitor. These monitors provide systolic, mean and diastolic blood pressure values by analyzing the pulse pressure oscillations through a cuff as it is occluded. Apparatus-specific algorithms were used to calculate the systolic and diastolic pressures, whereas the mean blood pressure was measured as the pressure where pulse pressure oscillations are highest. Oscillometric blood pressure monitors provide a better estimation of mean blood pressure in adult medium- to large-breed dogs under anesthesia compared to the Doppler technique [25]. Additionally, oscillometric BP monitoring was chosen because the nature of the procedures (orthopedic and dental), the clinical observational nature of this study and the health status of the patients (ASA type I-II) did not justify the invasive nature and added cost of direct arterial blood pressure monitoring. Several factors can influence the accuracy of blood pressure monitoring. Of note, the agreement between oscillometric BP and direct BP has been shown to vary depending on the blood pressure status (normotensive vs. hypotensive) of canine patients under anesthesia with acute hemorrhage [26]. In their study, oscillometric blood pressure overestimated the systolic blood pressure of hypotensive patients in comparison with the gold standard. How the etiology of hypotension affects the reliability of the indirect blood pressure readings and whether this also applies to volatile anesthetic-induced hypotension without acute hemorrhage is unclear, however. The indirect nature of blood pressure monitoring in this study may, however, represent a limit in the detection of hypotension in veterinary patients. Future studies should include direct blood pressure monitoring and more sophisticated means of assessing cardiovascular function such as cardiac output monitoring or perfusion index measurements. Patients enrolled in the current study may have had lower blood pressures than measured and the transiently exacerbated hypotension following administration of a fast IV infusion of a crystalloid may have been masked or missed.

Recent evidence also suggests that blood pressure measurements are poor predictors of hemodynamic response to fluid administration [16]. In hypotensive anesthetized patients, the response to a fluid bolus is variable. This is likely due to not only the effects of the anesthetics on contractility and vasomotor tone, but also the volume status of the individual patient. Adequate fluid load appears to be the most important determinant in the response to fluid therapy to improve blood pressure in dogs. However, excessive perianesthetic fluid therapy may be associated with worsened outcomes, including development of interstitial edema from fluid overload, decreased pulmonary functions resulting from increased lung water levels, reduced tissue oxygenation, increased risks of infection and delayed coagulation times [27,28]. Several indices have been developed to help guide therapeutic decision making in veterinary critical care and anesthesia. Pulse pressure variation is a useful tool to assess intravascular volume status and can be derived either from the arterial pressure waveform when direct blood pressure monitoring is available, or from plethysmography curves if arterial catheterization is not feasible [29].

## 5. Conclusions

In conclusion, the crystalloids used in the present study did not differ in their effect on BP and HR following rapid IV infusions in hypotensive anesthetized dogs. Future studies aimed at comparing different fluid therapy for inhalant anesthetic-induced hypotension in dogs would benefit from documenting continuous hemodynamic measures instead of intermittent blood pressure values.

## Figures and Tables

**Table 1 animals-12-01781-t001:** Ionic and buffering compositions of PLYTE, PLYTECA and LRS.

Fluid Type	Osmolality (mOsm L^−1^)	[Na^+^] (mEq L^−1^)	[K^+^] (mEq L^−1^)	[Cl^−^] (mEq L^−1^)	[Mg^2+^] (mEq L^−1^)	[Ca^2+^] (mEq L^−1^)	Lactate (mEq L^−1^)	Acetate (mEq L^−1^)	Gluconate (mEq L^−1^)
LRS	273	130	4	109	—	3	28	—	—
PLYTE	294	140	5	98	3	—	—	27	23
PLYTECA	294	140	5	98	1.5	—	—	27	23

**Table 2 animals-12-01781-t002:** Demographic data for dogs in each treatment group.

Treatment Group	LRS	PLYTE	PLYTECA
Number of dogs	13	9	8
Age (years)	2.8 (0.7–8.9)	2.3 (0.7–7.8)	3.5 (0.7–8.0)
Sex	6 MC, 7 FS	5 MC, 3 FS, 1F	5 MC, 3 FS
Mass (kg)	27.9 ± 9.8	36.2 ± 10.2	37.7 ± 7.0
Bolus fluid rate (mL r^−1^)	8.5 (5.0–10.0)	7.4 (4.7–9.9)	6.8 (4.9–8.6)

Data are mean (SD) for weight and median (range) for age and fluid rate. MC: male castrated, FS: female sterilized, F: female intact.

**Table 3 animals-12-01781-t003:** Effect of LRS, PLYTE and PLYTECA on MAP and HR of hypotensive dogs under isoflurane anesthesia.

Variable			Time Point (min)				
	Fluid Type	−5	0	5	10	15	20
MAP (mmHg)	LRS	61 ± 2.5	60 ± 4.1	60 ± 5.6	63 ± 5.4	63 ± 5.5	63 ± 6.9
	PLYTE	57 ± 7.0	61 ± 8.5	65 ± 7.1	62 ± 7.1	61 ± 8.3	66 ± 10.5
	PLYTECA	63 ± 6.3	66 ± 5.6	61 ± 5.6	60 ± 10.8	58 ± 11.5	66 ± 8.9
*p*-value				*p* = 0.07	*p* = 0.22	*p* = 0.15	*p* = 0.52
HR (beats per minute)	LRS	85 ± 18.8	84 ± 21.8	80 ± 15.7	81 ± 15.7	83 ± 17.7	85 ± 19.1
	PLYTE	92 ± 16.9	102 ± 18.7	94 ± 14.4	94 ± 18.4	92 ± 17.6	92 ± 17.7
	PLYTECA	89 ± 24.8	91 ± 25.2	93 ± 26.7	89 ± 24.4	95 ± 27.2	99 ± 28.4
*p*-value				*p* = 0.44	*p* = 0.61	*p* = 0.20	*p* = 0.14

Mean ± SD MAP and HR produced by three different isotonic crystalloid bolus infusions to hypotensive isoflurane-anesthetized dogs. Time points: −5: five minutes before onset of bolus; 0: onset of bolus; 5: 5 min after onset of bolus; 10: 10 min after onset of bolus; 15: 15 min after onset of bolus; 20: 20 min after onset of bolus.

## Data Availability

Raw data are stored in private computers, property of the University of Pennsylvania, School of Veterinary Medicine, and are available upon request.

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
