# Peer review of "Comparison of Mean Arterial Blood Pressure and Heart Rate Changes in Response to Three Different Randomized Isotonic Crystalloid Boluses in Hypotensive Anesthetized Dogs"

_animals, 2022, doi:10.3390/ani12141781_

Round 1

Reviewer 1 Report

Thank you for reporting the results of this very interesting study. As the author wrote at the beginning, there are many "anecdotal" methods out there, and I deeply agree with the study that demonstrates this.

I look forward to more studies like this to advocate that anecdotal methods should not be done haphazardly. The Introduction and Discussion are well written.

However, this study design is not perfect. Certainly MAP and HR are important in the assessment of circulating blood volume, but I get the impression that these alone are insufficient.

If possible, other measures of circulating blood volume and assessment of vascular volume by imaging studies would make this manuscript more compelling in regard to the results of the present study. If these have not been performed, then this should be added to the limitations. On the other hand, if you also measured systolic blood pressure, you could show the Shock index.

Also, you mention the effect of Mg+ in the infusion, but did you evaluate the serum Mg concentration before and after the procedure? Did you also evaluate electrolytes, blood osmolality, etc.? I believe this information is also important.

I have the impression that the animals in this study were not so severely injured, but the results may be different in higher-risk patients (hemorrhage, high-energy trauma, etc.). I think it is necessary to consider this in the future. I am particularly concerned about what happens in animals with electrolyte imbalance.

But on the other hand, I do not think that infusions are completely pointless. I am also very interested in whether there is a difference in the use of bolus infusions with or without boosting agents.

Please consider reconsidering the manuscript and adding limitations in light of the above.

Below are some specific comments.

Line105: Why are there differences in LRS infusion rates (2-5 ml/kg/hr)?

Author Response

Reviewer comment 1: "Thank you for reporting the results of this very interesting study. As the author wrote at the beginning, there are many "anecdotal" methods out there, and I deeply agree with the study that demonstrates this. I look forward to more studies like this to advocate that anecdotal methods should not be done haphazardly. The Introduction and Discussion are well written."

Answer 1: Thank you for your thoughtful consideration of our manuscript!

Reviewer comment 2: “However, this study design is not perfect. Certainly MAP and HR are important in the assessment of circulating blood volume, but I get the impression that these alone are insufficient. If possible, other measures of circulating blood volume and assessment of vascular volume by imaging studies would make this manuscript more compelling in regard to the results of the present study. If these have not been performed, then this should be added to the limitations. On the other hand, if you also measured systolic blood pressure, you could show the Shock index.”

Answer 2: Unfortunately, further evaluation of circulating blood volume was not performed in this study due to the observational nature of the study and the fact that these other techniques were not standard monitoring protocols at the institution where the study took place. We agree with the reviewer that it would have been interesting to assess cardiovascular function with variables beyond non-invasive blood pressure. If anything, this study paves the way to other prospective controlled studies evaluating what specific changes if any are brought about by the rapid administration of IV balanced isotonic crystalloid solutions during periods of hypotension in anesthetised dogs. We have added a mention of this limitation to the discussion as follow:

Line 327-338: “The indirect nature of blood pressure monitoring in this study may however represent a limit in the detection of hypotension in veterinary patients and their response to fluid therapy. Future studies should include direct blood pressure monitoring and more sophisticated means of assessing cardiovascular function such as cardiac output monitoring or perfusion index measurements.”

Reviewer comment 3: “Also, you mention the effect of Mg+ in the infusion, but did you evaluate the serum Mg concentration before and after the procedure? Did you also evaluate electrolytes, blood osmolality, etc.? I believe this information is also important.”

Answer 3: Thank you for this insightful comment. We did not perform serial serum magnesium concentration analysis, and cannot determine whether our interventions lead to acute changes in serum magnesium concentrations in patients. We thus modified the manuscript to reflect this as such: Line 249-252: “However, serial serum magnesium levels were not analyzed and the difference in Magnesium dose administered during a fluid bolus may not be sufficient to effect a significant change in a patient’s serum concentration let alone their blood pressures.”

Reviewer comment 4: “I have the impression that the animals in this study were not so severely injured, but the results may be different in higher-risk patients (hemorrhage, high-energy trauma, etc.). I think it is necessary to consider this in the future. I am particularly concerned about what happens in animals with electrolyte imbalance.

Answer 4: Your observation is correct: The patients in this study were healthy patients (ASA2) undergoing an elective orthopedic or dental anesthetised procedure. The results may very well differ when critically ill patients experience hypotension under general anesthesia as systemic inflammation and metabolic acidosis, among other factors, are associated with vasoplegia which may limit the benefit of fluid therapy for the treatment of isoflurane induced hypotension. We have therefore amended the current manuscript to clarify this limitation as follow:

Line 233-235: “The results of this study may not extend to patients with systemic illness or hypotension of different etiology.” 

Reviewer comment 5: “But on the other hand, I do not think that infusions are completely pointless. I am also very interested in whether there is a difference in the use of bolus infusions with or without boosting agents. Please consider reconsidering the manuscript and adding limitations in light of the above.”

Answer 5: We agree with the reviewer that infusions are not completely pointless. To this point, we argue that careful titration of anesthetic depth should be prioritized for treatment of anesthetic induced hypotension, as stated in lines 306 – 308:

 “It is possible that depth of anaesthesia may have been a more important contributor to the continued hypotension in the subjects of this study, rather than the initiation of a rapid IV fluid infusion.”

 Reviewer comment 6:

“Below are some specific comments. Line 105: Why are there differences in LRS infusion rates (2-5 ml/kg/hr)?”

Answer 6: The fluid rates described are within the recommended ranges in the institution where the study was performed. Different ranges may have been selected by different nurse anesthetists and were not standardized because of the observational nature of this study.

Reviewer 2 Report

Abstract
-Please define the abbreviations when you use them the first time.

Methods
-What is the dosage of propofol? Are there any differences between groups?
-Regional anesthesia causes different levels of hypotension, depending on the type. For example, epidural anesthesia can cause more hypotension than simple nerve block. Authors should detail the types of local blocks and analyze their differences.

Statistical analysis
- For non-normally distributed data, instead of repeated measures ANOVA, the Kruskal–Wallis test is a test for consecutive time data.

- Please describe all p values in the tables.
- The sample size is so small to indicate statistical significance. Further power analysis is required. 

Results
- Please move some sentences to the Method section and remove duplicate sentences.

- Demographic data seem to differ between groups. Differences in demographic data make the results less reliable. Therefore, further analysis will be required.

Discussion

- The result is very simple, but the discussion section seems to be very verbose and contains many unnecessary sentences.  It is recommended that the authors describe only the sentences that are directly related to the results obtained from the experiment.

-Line 217-218: "Ultimately, Mg2+ administration may lead to hypotension". According to Tables 1 and 2 (the results of this study), the content of Mg2+ in the Plasmalyte is 1.5-3 mEq/L, which is 1.8-3.6 mg/L.
In addition, according to the results of this study, the administrated content of Mg2+ is 0.0099 - 0.0228 mEq.  However, a study of reference 2 (Mg2+ induced hypotension) in which Mg2+ was administered at doses of 60, 90, and 120 mg kg-1. Therefore, the administered doses of Mg2+ do not appear to cause hypotension.

Author Response

Reviewer Comment #1:

“Abstract -Please define the abbreviations when you use them the first time.”

Answer 1: We have added definitions.

Reviewer Comment #2:
“Methods -What is the dosage of propofol? Are there any differences between groups?”

Answer #2: The total dose administered depended on the effect of the premedication and was always titrated slowly to effect, as per current standard operating procedures at the institution where the present observational study was performed. As stated, “The mean propofol dose required for induction of anaesthesia was 2.7 ± 1.8 mg Kg-1”. Differences in induction doses between groups were not evaluated. Onset of hypotension occurred at 25, 26 and 29 minutes from induction for LRS, PLYTE and PLYTECA groups respectively and the influence of induction agent dose on the response to treatment is not expected to be significant considering the short duration of effect of propofol and the fact that it was always carefully slowly titrated to effect.

Reviewer Comment #3:

“-Regional anesthesia causes different levels of hypotension, depending on the type. For example, epidural anesthesia can cause more hypotension than simple nerve block. Authors should detail the types of local blocks and analyze their differences.”

Answer #3: The specific type of block each patient received as part of its anesthetic protocol was not included in the analysis in the present study. Group allocation was performed at random so as to minimize the effect of specific anesthetic techniques on outcomes.

Reviewer comment #4:

“Statistical analysis - For non-normally distributed data, instead of repeated measures ANOVA, the Kruskal–Wallis test is a test for consecutive time data.

Answer #4: The Mood’s median test is a nonparametric test that is used to test the equality of medians from two or more populations. Therefore, it provides a nonparametric alternative to the one-way ANOVA. The Mood’s median test works when the Y variable is continuous, discrete-ordinal or discrete-count, and the X variable is discrete with two or more attributes.

Reviewer Comment #5: “- Please describe all p values in the tables.”

Answer: individual P-values for comparisons describing the effect of treatment after baseline are now included in the table.

Reviewer Comment #6:
- The sample size is so small to indicate statistical significance. Further power analysis is required. 

Answer: From the methods section, we stated that a power analysis was performed: “Despite the small number of patients included in this study (n = 30), a statistical analysis suggests that the sample size was sufficient to detect changes in blood pressure and HR equivalent to 20% and 35% from baseline, respectively at alpha = 0.05 and an 80% power.”.

Reviewer Comment #7:
“Results - Please move some sentences to the Method section and remove duplicate sentences”.

Answer #7: “The fluid rates used to deliver fluid boluses to each subject was limited by the maximum rate of the pumps (999 mL hr-1). Thus, patients weighing 25 kg and above received a maximum of 250 mL over a 15 min period. Fluid rates administered to patients less than 25 kg were limited to 10 mL Kg-1 over 15 min.”

was moved from Results section to Methods section.

Reviewer Comment #8:

“- Demographic data seem to differ between groups. Differences in demographic data make the results less reliable. Therefore, further analysis will be required.”

Answer 8: The only variable that was significantly different between groups was the body weight, which was significantly lower in the LRS group compared to the PLYTE and PLYTECA groups. This means fluid boluses were administered at a slower rate and potentially, the total fluid dosage administered in this group may have been larger than in the other groups were patients had a higher weight. This indeed complicates analysis of the data but this limitation is clearly described in the methods and results section. Further, the study is not sufficiently powered to perform analysis where groups are subcategorized in terms of their body weight. Future studies will take this confounding issue into consideration.

Reviewer Comment #9:

“Discussion - The result is very simple, but the discussion section seems to be very verbose and contains many unnecessary sentences.  It is recommended that the authors describe only the sentences that are directly related to the results obtained from the experiment.”

Answer to comment 9: Thank you for your critical evaluation of our discussion and its content. While we appreciate the importance of being concise, we believe each element in the discussion are important as they help contextualize the findings of the present study and they help highlight the limitations of our findings. We have reviewed the discussion’s syntax and modified the wording to shorten it, where appropriate.

Reviewer Comment #10:

-Line 217-218: "Ultimately, Mg2+ administration may lead to hypotension". According to Tables 1 and 2 (the results of this study), the content of Mg2+ in the Plasmalyte is 1.5-3 mEq/L, which is 1.8-3.6 mg/L.
In addition, according to the results of this study, the administrated content of Mg2+ is 0.0099 - 0.0228 mEq.  However, a study of reference 2 (Mg2+ induced hypotension) in which Mg2+ was administered at doses of 60, 90, and 120 mg kg-1. Therefore, the administered doses of Mg2+ do not appear to cause hypotension.

Answer to comment 10: Thank you for this thoughtful comment. We have modified the discussion to include a mention of the limited dose of Mg+ a patient would have received via a fluid bolus that reads as follow:

“ However, serial serum magnesium levels were not analyzed and the difference in Magnesium dose administered during a fluid bolus may not be sufficient to effect a significant change in a patient’s serum concentration let alone their blood pressures.”

Reviewer 3 Report

Animals June 2022 manuscript review –

Line 30-31: rephrase to state conclusion about effect on both MAP and HR

Line 32: rephrase to include the finding or conclusion of effect of LRS also

Line 80: seems to have typographical error – ‘then’ should be ‘than’

Line 122: typo error – ‘place’ should be ‘placed’

Line 166: the mean fluid rate values for LRS and PLYTECA differ from those listed in table 2. Please reconcile.

Line 225: place a comma after “Following a rapid IV fluid infusion

Author Response

Reviewer Comment #1: “Line 30-31: rephrase to state conclusion about effect on both MAP and HR”

Answer 1: We have rephrased the sentence as: “Intravenous isotonic crystalloid infusions over 15 minutes did not significantly change MAP or HR in hypotensive dogs under general anaesthesia.”

Reviewer Comment #2: Line 32: rephrase to include the finding or conclusion of effect of LRS also

Answer 2: We have rephrased the sentence as: “Neither LRS, PLYTE nor PLYTECA exacerbated hypotension or caused tachycardia.”

Reviewer Comment #3: Line 80: seems to have typographical error – ‘then’ should be ‘than’

Answer 3: We have corrected “then” for “than”

Reviewer Comment #4: Line 122: typo error – ‘place’ should be ‘placed’

Answer 4: We have corrected the typo as suggested.

Reviewer Comment #5: Line 166: the mean fluid rate values for LRS and PLYTECA differ from those listed in table 2. Please reconcile.

Answer 5: We have made corrections to the data presented and confirmed values from the original statistics output file.

Reviewer Comment #6: Line 225: place a comma after “Following a rapid IV fluid infusion

Answer 6: A comma was inserted in text as directed.

Round 2

Reviewer 2 Report

Thank you for your efforts to revise the paper. However, it seems that the “Discussion section” has not been revised.  Please write a simple and clear “Discussion section” based solely on your research findings. I recommend that you create a Limitation section to describe the limitations of your study.

Be consistent with the number of decimal places.